# The Role of the Cardiac Biomarkers in the Renal Cell Carcinoma Multidisciplinary Management

**DOI:** 10.3390/diagnostics13111912

**Published:** 2023-05-30

**Authors:** Anca Drăgan, Ioanel Sinescu

**Affiliations:** 1Department of Cardiovascular Anaesthesiology and Intensive Care, Prof. C.C. Iliescu Emergency Institute for Cardiovascular Diseases, 258 Fundeni Road, 022328 Bucharest, Romania; 2Department of Urological Surgery, Dialysis and Kidney Transplantation, Fundeni Clinical Institute, 258 Fundeni Road, 022328 Bucharest, Romania; urologiefundeni@gmail.com; 3Department of Uronephrology, Carol Davila University of Medicine and Pharmacy, 8 Eroii Sanitari Blvd, 050474 Bucharest, Romania

**Keywords:** renal cell carcinoma, cardio-oncology, risk assessment, global longitudinal strain, cardiac biomarkers

## Abstract

Renal cell carcinoma, an aggressive malignancy, is often incidentally diagnosed. The patient remains asymptomatic to the late stage of the disease, when the local or distant metastases are already present. Surgical treatment remains the choice for these patients, although the plan must adapt to the characteristics of the patients and the extension of the neoplasm. Systemic therapy is sometimes needed. It includes immunotherapy, target therapy, or both, with a high level of toxicity. Cardiac biomarkers have prognosis and monitoring values in this setting. Their role in postoperative identification of myocardial injury and heart failure already have been demonstrated, as well as their importance in preoperative evaluation from the cardiac point of view and the progression of renal cancer. The cardiac biomarkers are also part of the new cardio-oncologic approach to establishing and monitoring systemic therapy. They are complementary tests for assessment of the baseline toxicity risk and tools to guide therapy. The goal must be to continue the treatment as long as possible with the initiation and optimisation of the cardiological treatment. Cardiac atrial biomarkers are reported to have also antitumoral and anti-inflammatory properties. This review aims to present the role of cardiac biomarkers in the multidisciplinary management of renal cell carcinoma patients.

## 1. Introduction

Renal cell carcinoma (RCC) is reported as the ninth most common malignant neoplasm in the United States [1], where it has the highest incidence. Australia and New Zealand come next, followed by Europe [1]. Lifestyle and widespread imagistic investigations can explain the higher incidence of RCC in developed countries. During the last 2 decades until recently, there has been a 2% annual increase in the incidence worldwide and in Europe [2].

Smoking, obesity, and hypertension represent the established risk factors of this disease [2]. Type 2 diabetes was demonstrated to be associated with the risk of RCC in women [3]; yet, the Vital study did not find this association [4]. RCC is more frequent in males [2,4], with a higher incidence in the older population [2]. A bidirectional relationship exists between RCC and chronic kidney disease [5]. The two diseases have common risk factors. RCC can provoke chronic kidney disease via surgical reduction of renal mass and perioperative/medical therapies for acute kidney injury (AKI) [5], while chronic kidney disease can lead to RCC via an underlying cystic disease or oxidative stress [5].

Tumour Node Metastasis (TNM) classification remains the main criterion of treatment decisions, although the European Association of Urology (EAU) recently proposed a new RCC staging classification, a more clinically oriented TNM staging scheme, that can help clinicians decide the appropriate treatment based on the burden of the metastatic disease [6]. A great proportion of patients is RCC incidentally diagnosed during investigation of nonspecific symptoms. The RCC patient remains asymptomatic until the late stage of the disease [2]. About 30% of patients have already developed a metastasis or a locally advanced disease at the diagnosis time [7].

Although surgery remains the mainstay of the RCC treatment, systemic therapy is sometimes needed in metastatic disease. It includes immunotherapy, target therapy, or both, with a high level of toxicity. Studies such as CARMENA and SURTIME move the treatment of the advanced disease of cytoreductive nephrectomy to specific medical therapy [2,8].

The RCC patient management must be multidisciplinary. Well-known serum cardiac biomarkers, troponin I/T and the natriuretic peptides, have been studied increasingly in this setting. They help with the baseline assessment of the patient at the diagnostic time and in perioperative oncologic and nononcologic outcome evaluation of the patient. These serum biomarkers, complementary to other investigations, are involved in baseline assessment and monitoring the systemic medical treatment. They help the clinician to make a more personalised treatment choice. The global longitudinal strain (GLS), an imaging cardiac biomarker, is another important tool in oncologic patient evaluation. These cardiac biomarkers were recently introduced in the cardio-oncology guidelines issued by the European Society of Cardiology (ESC). This paper aims to present the role of cardiac biomarkers in the diagnosis, prediction, treatment optimisation, and personalisation of RCC treatment.

## 2. Methods

The present narrative review aims to present the role of cardiac biomarkers in the multidisciplinary approach in RCC treatment, combining the specific oncologic medical literature with the new issued 2022 ESC guidelines (perioperative assessment in noncardiac surgery and onco-cardiology) (Figure 1). We searched the online PubMed database using some keywords/phrases: “troponin in renal cell carcinoma”, “BNP in renal cell carcinoma”, “cardiac natriuretic peptide in renal cell carcinoma”, “myocarditis in renal cell carcinoma”, “global longitudinal strain in renal cell carcinoma”, “cardiac biomarkers in renal cell carcinoma”, “cardiac preoperative evaluation in renal cell carcinoma”, “myocardial injury in renal cell carcinoma”, “troponin all-cause mortality in oncology”, “cardiotoxicity in renal cell carcinoma”, “cardiotoxicity of immune checkpoint inhibitors”, “vascular endothelial growth factor inhibitors cardiotoxicity”, “anticancer effects of cardiac natriuretic peptides”. Abstracts of the English written articles were reviewed and selected upon determination of the relevancy of the subject. Case reports and case series were excluded. The selected full-length papers were further studied and evaluated upon determination of the relevancy, excluding the duplicates.

## 3. Cardiac Biomarkers

Troponin was first described in the 1960s as a third factor besides myosin and actin, which conferred calcium sensitivity to actomyosin [9,10]. Professor Ebashi proposed a scheme for the molecular mechanism of regulation of contraction and relaxation, in which troponin was described as a calcium receptive protein [11,12]. The three subunits of the troponin complex were discovered by Greaser and Gergely in 1971, with their specific properties: inhibition of adenosine-triphosphatase (ATPase) activity (troponin I), tropomyosin binding (troponin T), calcium binding capacity (troponin C) [9,13]. Later, the amino acid sequence and the gene involved were researched. The discovery of serum troponin revolutionised the diagnosis of myocardial infarction. Earlier discovered cardiac biomarkers (aspartate transaminase, lactate dehydrogenase, creatine kinase, myoglobin, and creatine kinase MB) provided low sensitivity in the first crucial hours [14].

Although all the three troponins are synthesised in the cardiac and the skeletal muscle, only troponins I and T, the cardiac troponins (cTn), are specific and sensitive to cardiac myocyte injury [15,16,17]. Because the amino acid sequences of cardiac troponin C and skeletal troponin C are identical, no assays have been developed for the troponin C component [18]. The blood detection of cTn is specific for cardiac injury [11,17]. Only 5–8% of cTn is free in the myocyte cytoplasm or part of an early releasable pool [19]. The rest is attached to the actin filaments in the cardiac sarcomere [15,19]. The cTn half-life in blood is about 2 h [19]. Potential mechanisms of troponin release have been proposed: myocyte necrosis, apoptosis, myocyte cell turnover, cellular release of proteolytic troponin degradation products, increased cellular wall permeability, and formation with the release of membranous blebs [19]. Thus, cTn are the preferred biomarkers for the evaluation of both acute (newly detected dynamic pattern) and chronic (persistently elevated) myocardial injury [16]. High-sensitivity cTn assays (hs-cTn) are recommended nowadays for routine clinical use [16]. These methods provided higher diagnostic accuracy at a low cost compared to the standard troponin assays [17]. Both hs-cTn T and I have comparable diagnostic accuracy, while hs-cTn T has greater prognostic accuracy [17].

In healthy adults, hs-cTn I and hs-cTn T values were higher in men than in women of the same age and increased after 55 years in both sexes [20,21]. The difference was higher in the hs-cTn I case [20,21]. Sepsis, hypovolemia, atrial fibrillation, pulmonary embolism, congestive heart failure, myocardial contusion, and renal failure are proven to be associated with a rise of troponin [22]. A large retrospective study showed that elevated cTn I was independently associated with a higher risk for cardiac and noncardiac death in cancer patients without any previous cardiac disease [23]. Higher values were associated with increasing death risk [23]. Another study demonstrated that hs-cTn T was an independent predictor of all-cause mortality in cancer patients, proposing hs-cTn T as a tool to identify high-risk patients earlier [24]. Pavo et al. demonstrated in a prospective study that hs-cTn T in patients with a cancer diagnosis has predicted mortality with a cut off of 0.005 ng/mL [25,26]. Elevated values were associated with the advanced tumour stage [25,26]. Troponins were studied in relationship with RCC, as well. A retrospective study reported that myocardial injury occurred in 38.8% and AKI in 42.7% of patients following nephrectomy with inferior vena cava thrombectomy [27]. Levels of cTn I were significantly higher in patients with these complications [27]. In general, troponin has been demonstrated to increase in systemic RCC treatment. The Javelin Renal 101 trial demonstrated that patients with high baseline cTn T receiving checkpoint inhibitor (ICI) and vascular endothelial growth factor receptor inhibitor (VEGFi) combinations need close monitoring for major adverse cardiovascular events (MACEs) [28]. ICI treatment is sometimes associated with myocarditis. Elevated troponin was found in 94% of such patients [26,29].

The natriuretic peptides (NPs) family-related but genetically distinct paracrine factors help regulate blood volume, blood pressure, ventricular hypertrophy, pulmonary hypertension, fat metabolism, and long bone growth. The natriuretic system mainly consists of atrial natriuretic peptide (ANP), B-type natriuretic peptide (BNP), and C-type natriuretic peptide (CNP) [30]. ANP is synthesised mainly by the atrial myocardium and BNP by the ventricular myocardium as a response to volume load and increased wall tension, and CNP is produced in endothelium [31]. ANP is synthesised from its precursor, pro-ANP, together with other atrial peptides: long-acting natriuretic peptide (LANP), kaliuretic peptide (KP), and vessel dilator (VD).

BNP was first purified in porcine brain by Sudoh et al. [32], but its higher concentration is at the cardiac level. The precursor of BNP, secreted by myocytes, is cleaved in BNP, the active amino acid and N-terminal pro-BNP (NT pro-BNP), an inert amino acid. BNP/NT pro-BNP represents the first investigation in suspected chronic heart failure, before echocardiography evaluation.

Guidelines stated that values of BNP < 35 pg/mL and NT pro-BNP < 125 pg/mL make heart failure unlikely [30]. BNP binds to BNP-receptors and is cleaved by endoproteases or excreted by kidneys, with a half-time of 20 min. NT pro-BNP is completely excreted by kidneys, with a half-time of 120 min [26]. In renal failure, NP values are increased, with NT pro BNP/BNP ratio inversely related to the glomerular filtration rate. Female sex, age, hypertension, diabetes, and atrial fibrillation have been associated with elevated NP levels [26]. In cancer, anaemia and fluid can influence NP values [26]. Studies demonstrated that BNP rises because of the cancer-related inflammation, while cancer and cancer therapy may contribute to cardiovascular (CV) diseases [25,26]. No significant association was found between the markedly elevated BNP values in cancer patients with comorbidities and the clinical evidence of volume overload or left ventricular dysfunction [33]. Papazisis et al. studied NT pro BNP as a predictive response in sunitinib treatment in metastatic renal cell carcinoma (mRCC) patients [34]. Tuñon et al. reported NT pro-BNP for the first time as an independent predictor of the appearance of malignancies in coronary artery disease patients [35].

GLS represents an imaging biomarker, measured by speckle-tracking echocardiography. It was proposed to be routinely used in preoperative evaluation of cancer patients to detect therapy-related cardiac dysfunction [36]. With a biological reproducibility of 6%, it is more sensitive in evaluating the LV contractility, myocardial deformation preceding the change of the left ventricle ejection fraction (LVEF) [36,37]. The SUCCOUR trial demonstrated GLS to be more precise than 2D echocardiography specifically in the oncology population [38]. A systematic review published in 2014 underlined the importance of myocardial strain in early detection of cardiotoxicity in patients during and after cancer chemotherapy [39].

A recent cardio-oncology guideline emerged by the ESC combines the serum biomarker (BNP, NT pro-BNP, cTn) with imaging biomarker GLS (part of the echocardiographic evaluation) in baseline assessment of oncologic patients [40]. Additionally, serum cardiac biomarkers were proposed to be a prognostic tool for cardiac outcomes in cancer patients and for the cancer-related mortality itself [23,41].

## 4. Serum and Imagistic Cardiac Biomarkers’ Role in Perioperative Evaluation of the RCC Patient

EAU underlines that surgery represents the curative treatment in localised RCC and introduces the old idea of risk stratification [2]. The treatment decision must be individualised, especially for frail patients, weighting the benefits against the high risk of perioperative complications and the risk of chronic kidney disease [2]. The need for a multidisciplinary approach has long been expressed in research papers in this framework [42,43]. The assessment of the functional capacity of the RCC patient, the evaluation of his cardiorespiratory disease severity, and the stage and degree of inferior vena cava involvement, together with the urgency of the intervention were previously proposed [43].

Nasrallah et al. proposed the PN-A4CH model (Age ≥ 75 years, American Society of Anesthesiologist class > 2, Anaemia, surgical Approach, Creatinine > 1.5 and history of Heart disease) as a risk index to predict MACE [44]. 

Other researchers have studied echocardiography as part of the preoperative evaluation of major noncardiac surgery, but found no benefit in low-risk patients, proposing BNP measurement in these cases [45]. A systematic review and meta-analysis from 2009 showed that elevated pre-operative BNP or NT pro-BNP measurement is a powerful, independent predictor of CV events in the first 30 days after noncardiac surgery [46]. Another systematic review reported that, comparing with a preoperative measurement alone, additional postoperative BNP or NT pro-BNP measurement enhanced risk stratification for death or nonfatal myocardial infarction at 30 days and ≥180 days after noncardiac surgery [47].

In 2022, ESC published a revised guideline concerning the preoperative evaluation in noncardiac surgery. This multidisciplinary assessment must take into consideration patient-related and surgery-related risks [48]. Major urological surgery is included among the intermediate-risk surgeries with a 1–5% risk of CV death, myocardial infarction, and stroke at 30 days [48]. If adrenalectomy is used, the surgery becomes high risk (>5%) [48]. The patient’s age, the presence or absence of CV risk factors (e.g., smoking, hypertension, diabetes, dyslipidaemia, family disposition), or already diagnosed CV disease and comorbidities represent the criteria exposed by the guideline concerning patient-related risk [48]. We can notice that smoking and hypertension are both CV risk factors and established risk factors for RCC [2,48]. This fact emphasises the importance of preoperative assessment of RCC patients. Compared to the old version of the guideline which stated that NT pro-BNP, BNP, and cTn may be considered for obtaining independent prognostic information for perioperative and late cardiac events only in high-risk patients, the recently revised ESC guideline increases the role of serum biomarkers in this setting. Thus, it is recommended to measure hs-cTn T or hs-cTn I before both intermediate- and high-risk noncardiac surgery, at 24 h and 48 h afterward, in patients who have known CV diseases, CV risk factors (including age ≥65 years), or symptoms suggestive of CV diseases [48]. From the BNP, NT pro-BNP point of view, the 2022 guideline suggests that these serum biomarkers should be measured in patients who have known CV diseases, CV risk factors (including age ≥65 years), or symptoms suggestive of CV diseases before intermediate- and high-risk noncardiac surgery [48].

Surgical treatment of RCC falls within these recommendations for many patients. Nephrectomies are associated with a high risk of complications in general, especially if we refer to AKI complications. Further, from this point of view, preoperative multidisciplinary assessment is mandatory [27,49]. Additionally, we must consider the results of other studies that report high values of serum cardiac biomarkers in oncologic patients, in general, and in RCC. One prospective study of 555 patients reported that NT pro-BNP, mid-regional pro-atrial natriuretic peptide (MR-pro ANP), mid-regional proadrenomedulin (MR-proADM), C-terminal proendothelin-1 (CT-pro-ET-1), and hs-cTn T were elevated in an unselected population of patients with cancer prior to induction of any cardiotoxic anticancer therapy [25]. The same markers and copeptin were strongly related to all-cause mortality, suggesting the presence of subclinical functional and morphological myocardial damage directly linked to disease progression [25]. A retrospective Japanese study of 2923 patients showed that BNP (66.4 ± 56.3 vs. 44.0 ± 35.3 pg/mL, *p* < 0.01) and C-reactive protein levels (0.99 ± 1.56 vs. 0.18 ± 0.27 mg/dL, *p* < 0.01) were significantly elevated in cancer patients comparing to the non-cancer patients, probably due to cancer-related inflammation [50]. BNP levels were significantly higher in the patients with stage IV cancer than in those with stage I, II, or III [50]. Kamai et al. published in 2018 research that found that preoperative serum levels of BNP and NT pro-BNP were related to the progression of RCC and a worse prognosis, their levels decreasing significantly after nephrectomy [51]. These serum biomarkers were not related to the expression of hypoxia-inducible factor 2 alpha (HIF-2 alpha) in the primary tumour or the serum level of vascular endothelial growth factor [51].

The authors draw attention to the possibility of subclinical functional and structural damage to the myocardium in advanced RCC [51]. Some researchers proposed a preoperative cardiac assessment algorithm that integrates clinical history, imaging, and serum biomarkers to provide a comprehensive assessment of the oncologic population [36]. They proposed GLS to be considered a routine in the preoperative cardiac assessment in this population because its value is more significant in the detection of asymptomatic LV dysfunction, especially with preserved LVEF [36]. Nowadays a multicentre, prospective study, PREOP-ECHO, is under way. Its aim is to assess the utility of preoperative echocardiography in patients undergoing intermediate- or high-risk noncardiac surgery, using classic echocardiographic parameters and GLS together with serum cardiac biomarkers [52]. Its recruitment is expected to be completed in June 2023 [52]. Recently, Kim et al., in a multicentric prospective study, demonstrated preoperative GLS to have an independent and incremental prognostic value in predicting early postoperative CV events and myocardial injury after noncardiac surgery [53]. Urology was among the studied surgeries.

A better preoperative assessment is needed also in metastatic RCC (mRCC). Flavin et al. highlight in 2016 the multidisciplinary approach in evaluating the opportunity of cytoreductive nephrectomy [54]. The mRCC patients are complex, with a high potential for major complications. The multidisciplinary team (surgeon, anaesthetist, oncologist) must carefully evaluate this type of patient, especially due to high probability of the intraoperative haemodynamic changes with arterial hypotension that can lead to a high rate of morbidity and mortality [54]. Mc Intosh et al. in 2020 found nine risk factors for increased risk of death to identify the patients less likely to benefit from cytoreductive nephrectomy [55]. Psutka reiterates the need for risk stratification, proposing a more personalised approach, for a careful selection of patients for whom cytoreductive nephrectomy is imperative for optimal outcomes, in an era of target therapy [8]. Although multiple active systemic treatments are now available for patients with metastatic kidney cancer, the tools to clearly identify those patients who may benefit from cytoreductive surgery remain poorly defined or validated [42].

## 5. Serum and Imagistic Cardiac Biomarkers’ Role in Systemic Therapy of RCC Patient

The place of systemic therapy in RCC treatment is explained by the current guidelines [2]. Risk assessment for mRCC proposed by EAU using the International Metastatic RCC Database Consortium (IMDC) risk model or Memorial Sloan-Kettering Cancer Center (MSKCC) score contains the Karnofsky performance to quantify the functional status of patients [2]. Yet, all these include no specific information about the cardiac status. The CardTox-Score, which contains clinical, GLS, serum biomarkers, and echocardiographic variables, represents a promising tool for predicting the risk of chemotherapy-induced cardiac toxicity in oncological patients undergoing non-anthracycline anticancer regimes, independently of the type of cancer [56], but it needs validation. The therapeutic choices for advanced/mRCC are target therapies, immunotherapy, or combinations [2]. A significant incidence of cardiotoxicity related to these treatments has already been reported.

Efforts have been made to predict better, quantify, and manage these complications. In 2022, the first guideline concerning cardio-oncology emerged from the ESC in order to help all the healthcare professionals providing care to oncology patients before, during, and after their cancer treatments with respect to their CV health and wellness [40]. To provide optimum prevention of the CV risk factors and diseases, all patients must have, before starting any systemic oncologic treatment, a baseline assessment without delaying oncological treatment. The baseline risk stratification recommended by the ESC cardio-oncology guideline is the Heart Failure Association-International Cardio-Oncology Society (HFA-ICOS) methodology [40]. The evaluation is a personalised, multidisciplinary process, which includes checking the clinical aspects (cancer treatment history, CV history and risk factors, physical examination, vital signs measurements) and complementary tests (BNP or NT pro-BNP, cTn, electrocardiogram, fasting plasma glucose/haemoglobin A1C, kidney function, lipid profile, transthoracic echocardiogram) [40]. Cardinale et al. explain that the release is the result of the myocardial cell damage, secondary to anti-cancer therapy. This phenomenon can lead to myocardial deformation with a decrease of GLS in the asymptomatic cardiotoxicity stage, followed by symptomatic heart failure [57].

The onco-cardiology ESC guideline recommends the baseline measurement of BNP/NT pro-BNP and/or cTn in all patients with cancer at risk of cancer therapy-related cardiac dysfunction (CTRCD) if these biomarkers are going to be measured during treatment [40]. The increased levels of these serum biomarkers have been defined as cTnI/T > 99th percentile, BNP ≥ 35 pg/mL, and NT pro-BNP ≥ 125 pg/mL [40].

Transthoracic echocardiography (TTE) was recommended as the first-line modality for the assessment of cardiac function in patients with cancer. Three-dimensional echocardiography was preferred as the echocardiographic modality to measure LVEF, while GLS was recommended in all patients with cancer having echocardiography, if available [40]. GLS evaluation becomes very important in patients with low–normal LVEF to confirm or not asymptomatic damage [58]. The 15% threshold comparing to baseline was reported to maximise specificity and minimise overdiagnosis of oncologic toxicity CV diseases and guide cardioprotective therapy [40,59]. The potential role of GLS in cardiac and oncological patients undergoing cardio-oncology rehabilitation is under analysis [60]. Researchers recommended using the same vendor to analyse GLS during cancer treatment to compare values over time more accurately [61]. Liu et al. provided some tips to obtain accurate and reproducible data in GLS measurement in onco-cardiology [58]. LVEF, GLS, and serum biomarkers define the three degrees of CTRCD in asymptomatic patients (Table 1).

VEGFi treatments are demonstrated to be associated with a wide array of CV complications, including hypertension, heart failure, QTc prolongation, and acute vascular events [40]. The calculated relative risk of mortality in a meta-analysis was 2.23 (CI 95%: 1.22–4.44) in patients who received sunitinib, sorafenib, or pazopanib [63]. The patients who may have a potential benefit from VEGFi treatment are oncologic patients with advanced disease. In their case, the goal must be to continue the systemic specific treatment for as long as possible with initiation or optimisation of CV treatment. Ischemic cardiac events have been reported in 3% of patients treated with sorafenib [64], in 4% of patients treated with sunitinib [65], and in 2% of patients with pazopanib [65]. Cardiac dysfunction has been found in 11% and 13% of patients treated with sunitinib, respective of pazopanib [65]. A significant decline of LVEF from the baseline was demonstrated in 1.4% of sorafenib patients and in 1.8% of sunitinib patients [66]. Sunitinib was shown to determine symptomatic heart failure in 15% of patients [67]. Catino et al., in a multicentre longitudinal prospective cohort study, found the worsening of the left diastolic function and the filling pressure in patients treated with sunitinib measuring E/e’ ratio and demonstrated the decline of the LVEF of 9.7% (LVEF decline ≥10% to a value <50%) in the first cycle of treatment, during the first 3.5 weeks after initial sunitinib treatment [68]. Hypertension is the most frequent CV complication of cabozantinib, the risk of LV systolic dysfunction being only modest [69,70]. Pazopanib treatment was associated with a 40% incidence of hypertension and 26% elevation of NT pro-BNP, although only 2.4% cases of heart failure were reported [69,71,72]. Hall et al. studied 159 patients who received targeted therapies for mRCC. Asymptomatic cardiotoxicity, defined by an elevated NT pro-BNP level and/or a decrease in systolic function as estimated by LVEF, was identified in 27% patients [72]. Sunitinib was the agent most frequently used, with 65% of sunitinib-treated patients developing a form of CV toxicity, excluding hypertension [72]. Other VEGFi, bevacizumab, sorafenib, and pazopanib, also elicited significant CV toxicity, with incidences ranging from 51% to 68% [72].

VEGFi treatments (pazopanib, sunitinib, sorafenib) can cause left ventricular systolic dysfunction through direct and indirect mechanisms, causing coronary artery disease and hypertension [73]. CV toxicity is often reversible at VEGFi treatment stop [73]. That is why it is important to monitor the patients in order to early detect and treat the CV side effects, before these become irreversible. ESC cardio-oncology guidelines provided the recommendations for baseline evaluation and surveillance in patients with VEGFi treatment (Table 2).

Cardiotoxicity related to ICI is reported by studies at 1% [74,75], 3.1% [74,76], and 9.7% [55,58] due to misclassification of CV disease [74]. The most severe cardiac complication of ICI treatment is myocarditis with a high fatality rate. It may develop during the first 12 weeks of treatment, although late cases (after week 20) may occur [40,77]. Other side effects include dyslipidaemia, myocardial infarction, vasculitis, atrioventricular block, supraventricular and ventricular arrhythmias, sudden death, Takotsubo-like syndrome, non-inflammatory heart failure, pericarditis, pericardial effusion, and ischaemic stroke [40]. Salem et al. reported in a large case series of 122 patients the early onset of symptoms (median of 30 days after initial exposure to ICI) and up to 50% mortality in ICI-associated myocarditis [78]. In high-risk patients and in patients with high baseline cTn levels, TTE monitoring may be considered [40]. According to the guideline, high baseline ICI-related CV toxicity risks include ICI dual therapy, combination of ICI with others cardiotoxic therapies, patients with ICI-related non-CV events, or prior CTRCD or CV disease [40]. ESC cardio-oncology guideline recommendations of ICI CV assessment are presented in Table 3.

The Javelin Renal 101 trial studying patients receiving ICI and VEGFi combinations, recommended close monitoring of the patients with high baseline cTn T [28]. When myocarditis is suspected, new biomarkers changes, or new cardiac symptoms occur, cardio-oncology evaluation is strongly recommended [40]. Although the guideline stated that both cTn I and cTn T can be used in ICI myocarditis diagnostic [40], cTn I seems to be more useful, because cTn T may be elevated with myositis [40,74]. NP are less specific for the myocarditis diagnostic [74]. The criteria of the clinical diagnosis of ICI-related myocarditis include cTn elevation (new or significant change from baseline) and one major criterion (cardiac magnetic resonance diagnostic-modified Lake Louise criteria) or two minor criteria (clinical syndrome, ventricular arrhythmia, decline in LV systolic function, suggestive cardiac magnetic resonance, other immune-related adverse events) [40]. GLS may detect early changes in left contractile function in this setting [61].

## 6. Cardiac Natriuretic Peptides—A Potential Therapeutic Target in RCC

Cardiac natriuretic peptides not only have monitoring value in RCC management, but they also represent a novel therapeutic modality [31,79]. Atrial NPs can induce the death of cancer cells without significantly affecting normal cells [80]. Vesely et al. demonstrated that the number of human renal carcinoma cells in vitro was significantly reduced by KP, VD, and LANP [81,82,83]. When an increased concentration of KP, ANP, and LANP of 100 μM was used, the number of renal cancer cells significantly decreased by 70–74% within 24 h [81]. ANP, KP, VD, LANP, and urodilatin, a renal NP, have potent anticancer effects by eliminating up to 81% of renal carcinoma cells within 24 h of treatment [81]. ANP, KP, VD, and LANP have anticancer effects in vitro when given in concentrations above those normally circulating in the human body [31,82]. BNP does not have these properties [31], although a few reports propose also BNP in cancer treatment [84].

Zhang et al. described several signalling pathways that impact the progression of RCC, including VHL-HIF-VEGF angiogenesis signalling, PI3K/AKT/mTOR signalling, epithelial-to-mesenchymal transition-related pathways, and the Wnt/β-catenin pathway [85]. These mechanisms conduct the RCC growth, invasiveness, metastasis, and angiogenesis. 

The antiproliferative effect of ANP is mainly related to its interaction with the natriuretic peptide receptors A, B, and C (NPR-A, NPR-B, NPR-C). NPR-C is meanly the clearance receptor of NP, while the other two are guanylyl cyclase-linked receptors that mediate the antiproliferative effect of ANP [31]. ANP functions as a multikinase inhibitor, inhibiting some metabolic targets, including the Ras-MEK1⁄2-ERK1⁄2 kinase cascade. This NP also impacts the Wnt/β-catenin pathway and the pH regulation ability of cancer cells through a Frizzled-related mechanism [86,87]. Atrial NPs can block VEGF-induced endothelial cell proliferation [88] and inhibit VEGF and VEGF receptor 2 in human cancer cell lines [89]. Skelton et al. reported that VD, KP, ANP, and LANP maximally reduced the concentration of AKT by 31%, 32%, 31%, and 31%, respectively, in renal adenocarcinoma cells (*p* < 0.001) [90]. VD, LANP, KP, and ANP are also potent inhibitors of c-Fos and c-Jun proto-oncogenes within the nucleus of cancer cells. Over a concentration range of 100 pM–10 μM, these NP can maximally decrease c-Fos by 82%, 73%, 78%, and 74%, respectively, and c-Jun by 47%, 43%, 57%, and 49%, respectively, in renal cancer cells [91].

Cardiac hormones (ANP, KP, VD, LANP) can modify the balance of the pH value of the tumour extracellular microenvironment [92]. The Na/K exchanger isoform 1 (NHE-1) is activated by ANP, KP, VD, and LANP to increase the cancer cells’ intracellular acidity and further inhibit Wnt/β-catenin signalling [31,92].

Atrial NPs were reported to inhibit perioperative systemic inflammation and cancer recurrence [93]. Nojiri et al. reported in a large observational study that the perioperative low-dose atrial NPs reduced the inflammatory response to surgical trauma and postoperative cardiopulmonary complications in lung cancer surgery [94].

Studies reported that atrial NPs have antitumor and anti-inflammatory properties [93]. That is why the development of new anticancer agents based on the atrial NPs formula was desired. This method can reduce drug resistance and toxicity. Yet, new technologies are needed to improve stability and prolong the duration of their action. Xu et al. propose two possible methods: the virus packaging format and the use of these peptides in conjunction with other drugs [31]. The latter method is named the Tandem expression technique, which creates new peptides by fusion with a backbone protein to enhance the stability of NPs [31].

## 7. Future Directions

Recently, a lot of research focused on optimising RCC patient management perioperatively and during the specific systemic therapy. New guidelines appeared, and others have been updated. Thus, the need for a multidisciplinary and personalised approach in RCC becomes important. Serum (troponin, BNP, NT pro-BNP) and imaging (GLS) biomarkers are increasingly studied in this setting, and, now, they are part of the guidelines recommendations. The CardTox-Score, which contains clinical, GLS, serum biomarkers, and echocardiographic variables, represents a promising tool for predicting CTRCD risk in oncological patients undergoing non-anthracycline anticancer regimes, independently of the type of cancer, but it needs validation. At the same time, new biomarkers have to be developed and studied both from the cardiac and the oncological point of view. Novel biomarkers (myeloperoxidase, C-reactive protein, galectin-3, arginine–nitric oxide metabolites, growth differentiation factor-15, placental growth factor, fms-like tyrosine kinase-1, micro-ribonucleic acids, and immunoglobulin E.6) are searched in this setting, although the guideline did not support their routine measurement yet. Another future direction may be related to the therapeutic, antiproliferative, and anti-inflammatory value of the atrial natriuretic peptides, not only at the laboratory level, but also in clinical practice. Considering the large amount of data that needs to be analysed in order to personalise and optimise RCC treatment, new tools have to be developed, including artificial intelligence.

## Figures and Tables

**Figure 1 diagnostics-13-01912-f001:**
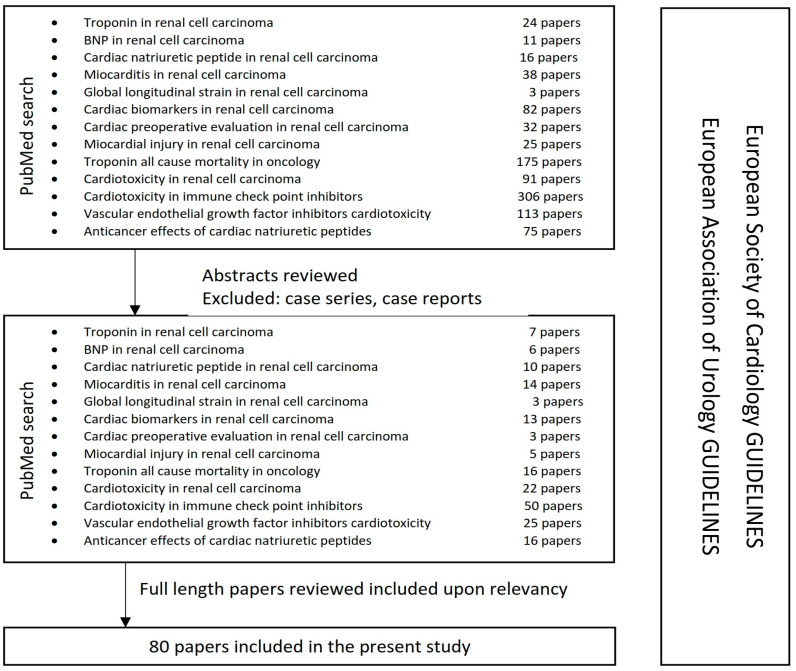
The methodology used by the authors.

**Table 1 diagnostics-13-01912-t001:** The degree of asymptomatic cancer therapy-related cardiac dysfunction.

**Asymptomatic** **CTRCD**	**LVEF**	**GLS and Serum Biomarkers**
**mild**	LVEF ≥ 50% [40,62]	new relative decline by >15% from the baseline And/Or a new increase of serum biomarkers [40,62]
**medium**	LVEF of 40–49% or a new LVEF decline by ≥10% [40,62]	
	new LVEF decline by <10% to LVEF 40–49% [40,62]	a new relative decline by >15% of GLS Or a new increase of serum biomarkers [40,62]
**severe**	new reduction of LVEF to <40% [40,62]	

Abbreviations: CTRCD: cancer therapy-related cardiac dysfunction; GLS: global longitudinal strain; LVEF: left ventricular ejection fraction.

**Table 2 diagnostics-13-01912-t002:** Recommendations for cardiac assessment in VEGFi treatment. Adapted from Lyon et al. (2022) [40].

HFA-ICOS Risk Stratification	Baseline	Surveillance
Low risk	○ clinical evaluation ○ BP measurement ○ ECG ○ TTE	○ clinical evaluation ○ BP measurement is recommended at every clinical visit ○ daily monitoring of BP during the first cycle and after each increase of the VEGFi dose and every 2–3 weeks thereafter
Moderate risk	○ clinical evaluation ○ BP measurement ○ ECG ○ TTE ○ BNP/NT pro-BNP	○ clinical evaluation ○ BP measurement is recommended at every clinical visit ○ daily monitoring of BP during the first cycle and after each increase of the VEGFi dose and every 2–3 weeks thereafter ○ TTE may be considered every 4 months during the first year and every 6–12 months in long-term treatment ○ BNP/NT pro-BNP may be considered every 4 months in the first year
High and very high risk	○ clinical evaluation ○ BP measurement ○ ECG ○ TTE ○ BNP/NT pro-BNP	○ clinical evaluation ○ BP measurement is recommended at every clinical visit ○ daily monitoring of BP during the first cycle and after each increase of the VEGFi dose and every 2–3 weeks thereafter ○ consider an ECG 2 weeks after starting treatment and in the case of dose increase ○ QTc monitoring is recommended monthly during the 3 months and 3–6 months thereafter ○ TTE should be considered every 3 months in the first year and every 6–12 months in long-term treatment ○ BNP/NT pro-BNP should be considered at 4 weeks after starting the treatment (in very high-risk patients together with TTE) and every 3 months during the first year ○ if BNP/NT pro BNP are not available, an additional TTE should be considered at 4 weeks after starting the treatment in selected patients

Abbreviations: BNP: B-type natriuretic peptide; BP: blood pressure; ECG: electrocardiogram; HFA-ICOS: Heart Failure Association-International Cardio-Oncology Society; NT pro-BNP: N-Terminal Pro-B-type natriuretic peptide; QTc: corrected QT interval; TTE: transthoracic echocardiography; VEGFi: vascular endothelial growth factor inhibitors.

**Table 3 diagnostics-13-01912-t003:** Recommendations for cardiac assessment in ICI treatment. Adapted from Lyon et al. (2022) [40].

	Before Starting Therapy	Surveillance
Low-risk patients	▪ CV ^1^ assessment, cTn are recommended ▪ TTE may be considered	CV ^1^ assessment may be considered every 6–12 months. Serial ECG and cTn before doses 2, 3, 4. If normal, reduce to every 3 doses until completion of the therapy.
High-risk patients ^2^	▪ CV ^1^ assessment, cTn, TTE are recommended	CV ^1^ assessment is recommended every 6–12 months in long-term treatment, may be considered in rest. Serial ECG and cTn before doses 2, 3, 4. If normal, reduce to every 3 doses until completion of the therapy.

^1^ CV assessment: physical examination, BP, BNP or NT pro BNP, lipid profile, HbA1c, ECG. ^2^ High-risk patients: dual ICI, combination ICI-cardiotoxic therapy, ICI related non-CV events, prior cardiotoxicity or CV disease. Abbreviations: BNP: B-type natriuretic peptide; BP: blood pressure; CV: cardiovascular; cTn: cardiac troponin; ECG: electrocardiogram; HbA1C: haemoglobin A1C; ICI: immune checkpoint inhibitors; NT pro-BNP: N-Terminal Pro-B-type natriuretic peptide; TTE: transthoracic echocardiography.

## Data Availability

Not applicable.

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
