# Peer review of "The Role of the Cardiac Biomarkers in the Renal Cell Carcinoma Multidisciplinary Management"

_diagnostics, 2023, doi:10.3390/diagnostics13111912_

Round 1

Reviewer 1 Report

This is an original review because today cardio oncology is a new branch of oncology. I think it is a useful work for clinicians who must evaluate possible cardiac toxicities related to antineoplastic treatments. In clinical practice cardiac biomarkers are not so used for prevention of toxicities in patients at risk, but to monitor the specific treatment of the toxicity that occurred during the treatment.

The only thing I would remove as it doesn't seem useful in the speech is the introduction on surgery in chapter 3 ,as it is not so pertinent in the speech.

Reviewer 2 Report

The manuscript by Dragan and Sinescu is an extensive review of the literature about the use of cardiac biomarkers in the multidisciplinary management of renal cell carcinoma patients.

Although I have some specific objections:

1.      The manuscript title should be changed, e.g. “The role of cardiac ………in …………”

2.      The manuscript has been divided into several sections, but in my opinion the section containing limitations of the cardiac biomarkers as potential therapeutic target in RCC should be added (e.g. before Future directions).

3.      Authors should describe how they searched for the references on which the manuscript was based. It can be presented e.g. in a Figure.

4.      Some sentences need stylistic correction – please check.

5.      Tables present very important results, but it would be better to present one table in one page, not one table in two pages.

6.      Some sentences contain redundant spaces, e.g. line 57 [2, 8] – should be [2,8].

Some sentences need stylistic correction – please check.

Reviewer 3 Report

The article appears to be of little scientific interest and the topic is not innovative. Overall it does not appear to be suitable for publication in this journal.

I suggest moderate editing of English language.

Round 2

Reviewer 2 Report

The manuscript has been significantly improved.